# Genomics of Gulf War Illness in U.S. Veterans Who Served during the 1990–1991 Persian Gulf War: Methods and Rationale for Veterans Affairs Cooperative Study #2006

**DOI:** 10.3390/brainsci11070845

**Published:** 2021-06-25

**Authors:** Krishnan Radhakrishnan, Elizabeth R. Hauser, Renato Polimanti, Drew A. Helmer, Dawn Provenzale, Rebecca B. McNeil, Alysia Maffucci, Rachel Quaden, Hongyu Zhao, Stacey B. Whitbourne, Kelly M. Harrington, Jacqueline Vahey, Joel Gelernter, Daniel F. Levey, Grant D. Huang, John Michael Gaziano, John Concato, Mihaela Aslan

**Affiliations:** 1Center for Behavioral Health Statistics and Quality, Substance Abuse and Mental Health Services Administration, Rockville, MD 20857, USA; kradhakrishnan05@gmail.com; 2College of Medicine, University of Kentucky, Lexington, KY 40536, USA; 3VA Cooperative Studies Program Epidemiology Center—Durham, Department of Veterans Affairs, Durham, NC 27705, USA; elizabeth.hauser@duke.edu (E.R.H.); dawn.provenzale@va.gov (D.P.); rmcneil@rti.org (R.B.M.); jacqueline.vahey@va.gov (J.V.); 4Department of Biostatistics and Bioinformatics, Duke University, Durham, NC 27705, USA; 5Cooperative Studies Program Clinical Epidemiology Research Center (CSP-CERC), VA Connecticut Healthcare System, West Haven, CT 06516, USA; renato.polimanti@yale.edu (R.P.); alysia.maffuci@va.gov (A.M.); hongyu.zhao@yale.edu (H.Z.); joel.gelernter@yale.edu (J.G.); mihaela.aslan@va.gov (M.A.); 6School of Medicine, Yale University, New Haven, CT 06511, USA; john.concato@yale.edu; 7Center for Innovations in Quality, Effectiveness, and Safety (IQuESt), Michael E. DeBakey VA Medical Center, Houston, TX 77030, USA; 8Department of Medicine, Baylor College of Medicine, Houston, TX 77030, USA; 9Massachusetts Veterans Epidemiology Research and Information Center (MAVERIC), VA Boston Healthcare System, Boston, MA 02130, USA; rachel.quaden@va.gov (R.Q.); stacey.whitbourne@va.gov (S.B.W.); kelly.harrington@va.gov (K.M.H.); michael.gaziano@va.gov (J.M.G.); 10Department of Biostatistics, Yale School of Public Health, New Haven, CT 06520, USA; 11Department of Medicine, Harvard Medical School, Boston, MA 02115, USA; 12Department of Medicine, Division of Aging, Brigham and Women’s Hospital, Boston, MA 02115, USA; 13Department of Psychiatry, Boston University School of Medicine, Boston, MA 02118, USA; 14Computational Biology and Bioinformatics Program, Duke University, Durham, NC 27705, USA; 15Division of Human Genetics, Department of Psychiatry, School of Medicine, Yale University, New Haven, CT 06511, USA; daniel.levey@yale.edu; 16Department of Psychiatry, Veterans Affairs Connecticut Healthcare Center, West Haven, CT 06516, USA; 17Cooperative Studies Program, VA Office of Research and Development, Washington, DC 20420, USA; grant.huang@va.gov; 18Food and Drug Administration, Silver Spring, MD 20993, USA

**Keywords:** Persian Gulf War deployment status, Gulf War Illness, phenotyping, genomics, exposures, U.S. veteran

## Abstract

Background: Approximately 697,000 members of the U.S. Armed Forces were deployed to the Persian Gulf in support of the 1990–1991 Persian Gulf War (GW). Subsequently, many deployed and some non-deployed veterans developed a chronic multi-symptom illness, now named Gulf War Illness (GWI). This manuscript outlines the methods and rationale for studying the genomics of GWI within the Million Veteran Program (MVP), a VA-based national research program that has linked medical records, surveys, and genomic data, enabling genome-wide association studies (GWASs). Methods: MVP participants who served in the military during the GW era were contacted by mail and invited to participate in the GWI study. A structured health questionnaire, based on a previously tested instrument, was also included in the mailing. Data on deployment locations and exposures, symptoms associated with GWI, clinical diagnoses, personal habits, and health care utilization were collected. Self-reported data will be augmented with chart reviews and structured international classification of disease codes, to classify participants by GWI case status. We will develop a phenotyping algorithm, based on two commonly used case definitions, to determine GWI status, and then conduct a nested case-control GWAS. Genetic variants associated with GWI will be investigated, and gene–gene and gene–environment interactions studied. The genetic overlap of GWI with, and causative mechanisms linking this illness to, other health conditions and the effects of genomic regulatory mechanisms on GWI risk will also be explored. Conclusions: The proposed initial GWAS described in this report will investigate the genomic underpinnings of GWI with a large sample size and state-of-the-art genomic analyses and phenotyping. The data generated will provide a rich and expansive foundation on which to build additional analyses.

## 1. Introduction

In response to Iraq’s invasion of Kuwait on 2 August 1990, a multinational coalition force was created to liberate Kuwait [1]. Approximately 697,000 members of the U.S. Armed Forces, mainly men (approximately 7% women [2]), were deployed in support of the 1990–1991 Persian Gulf War [1,3,4]. Although the duration of deployment was brief, and rates of injury and disease were low compared to other wars, a substantial fraction of veterans reported a chronic multi-symptom illness in temporal association with deployment [1,5,6]. For example, in an army base in Indiana, 125 Gulf War veterans presented in early 1992 with various symptoms. The most common symptom was fatigue, reported by 71% of the 79 soldiers who were subsequently evaluated by a multidisciplinary team [7]. Other symptoms endorsed by a majority of the 79 included sleep disturbance (57%), forgetfulness (54%), and joint pain (54%). Despite thorough medical and psychiatric evaluations and numerous screening tests, no unifying diagnosis could be made, and the symptoms were attributed to stress [7].

Deployment to the Persian Gulf was inherently stressful, and military personnel were also exposed to agents of potential concern, identified over the ensuing years, such as pyridostigmine bromide, infections, pesticides, solvents, depleted uranium, and air pollutants from burning trash and oil well fires [1,3,4,6,8]. In addition, both deployed and non-deployed veterans received multiple vaccinations in a short time frame. It has been suggested that some of these exposures could be related to risk for Gulf War Illness, based on self-reports, but this hypothesis is not supported when vaccination data based on medical records are considered [9]. In response to concerns regarding illness attributed by veterans to Gulf War service, various clinical and research programs were initiated by the Department of Defense and the Department of Veterans Affairs, and a workshop was held in April 1994 specifically to explore the evidence for increased incidence of unexpected illnesses attributable to deployment to the Persian Gulf [10]. The general conclusion was that exposure to the Persian Gulf theater of operations had produced adverse health effects, but no single disease or syndrome was apparent, and further research was necessary to better characterize this condition [1,3,8,10,11].

Numerous clinical and epidemiologic studies have since been undertaken in various coalition nations to understand the etiology, pathophysiology, and prognosis of what is now called Gulf War Illness (GWI) [12,13,14], and to develop a reliable and clinically useful case definition [8,14,15,16,17,18,19,20,21,22]. GWI remains a symptom-based illness, as defined by two generally accepted research case definitions, referred to as CDC [23] and Kansas [24]. Both definitions are based on self-reported chronic symptoms (>6 months in duration): in the CDC classification involving any two of three domains (fatigue, musculoskeletal pain, cognitive/mood) [23]; and in the Kansas definition involving any three of six domains (fatigue, musculoskeletal pain, cognitive/neuro/mood, pulmonary, gastrointestinal, skin) [24]. In addition, the latter definition considers exclusionary criteria (e.g., serious mental illness, multiple sclerosis, diabetes) that might explain the symptom complex in an individual [24]. Despite more than $300 million being invested by federal agencies in GWI research since 1992 [25,26,27,28], the causative agent(s) and the underlying pathophysiology of GWI remain elusive, as have effective treatments [14]. It is clear, however, that this condition affects a large fraction of deployed, and a smaller fraction of non-deployed, military personnel [8,14,15,22,23,24,29,30,31,32,33,34,35,36,37,38,39,40,41,42,43,44,45,46,47,48,49].

Given the complex heterogeneous nature of GWI and the long latency period in some veterans but not in others, genetic and epigenetic studies have the potential to improve understanding of this condition and possibly identify biologic mechanisms of disease [8,14,16]. Interactions between genetic and environmental factors, such as exposure to toxic chemicals, may play a role in the underlying pathophysiology [14,17,22]. For example, genetic variants of the xenobiotic metabolizing enzymes paraoxonases (PON) and butyrylcholinesterase (BChE) have been examined in a number of studies involving Gulf War era veterans, and inferences made concerning their associations with GWI [12,14,17,22,50]. However, based on a critical analysis of several genetic studies, the Institute of Medicine (IOM) concluded that these studies were underpowered and inconsistent [12,14]. Large-scale genome-wide studies are needed to dissect the polygenic architecture of complex diseases and traits [51], including multi-factorial disorders such as GWI.

To expand the scope of ongoing epidemiologic and biomarker research on GWI, the Department of Veterans Affairs (VA) initiated the VA Cooperative Studies Program (CSP) #2006, “Genomics of Gulf War Illness in Veterans”, to conduct a case-control, genome-wide association study (GWAS) in a large cohort of Gulf War era veterans. CSP #2006 is linked to the VA Million Veteran Program (MVP), a VA-based infrastructure resource for conducting genomic research [52]. Designed to facilitate the study of how genes affect health, MVP is developing a large database of genotypes and health information from the VA’s electronic health record (EHR) system and surveys collected from Veterans Health Administration (VHA) enrollees. The VA has a long-established history of successfully using EHR systems in research [53], thus providing us with an unparalleled opportunity to link participants’ genetic data to clinical outcomes. MVP enrollees include veterans who served in the military during the Persian Gulf War, and CSP #2006 will analyze data from MVP for this subgroup.

The primary objective of CSP #2006 is to identify genetic variants associated with GWI. A secondary objective is to examine interactions between genetic variants and self-reported Gulf War environmental exposures on risk of developing GWI. We will also examine (i) the genetic overlap of GWI with other physical and mental disorders, (ii) potential causative mechanisms that may link GWI to other health conditions, and (iii) effects of genomic regulatory mechanisms across tissues and cell types on GWI risk. These investigations will provide an unprecedented opportunity to classify and understand GWI. In this paper, we report on several aspects of CSP #2006, focusing on the initial planned genomic analyses.

## 2. Methods

### 2.1. Overview

The primary analysis of CSP #2006, and the focus of this report, is a GWAS of GWI. We will conduct a case-control study nested within the cohort of 1990–1991 Persian Gulf War era veterans in the MVP sample. We will select cases with GWI and controls without GWI from the genotyped MVP participants who served during this war, irrespective of deployment status. The preliminary designation of GWI cases will be based on two commonly used definitions, both endorsed by the IOM [13] (as described below). The pool of potential GWI cases will be identified using a phenotyping algorithm that primarily evaluates self-reported information from a questionnaire mailed to participants. Case status from self-reported data will be corroborated with VA’s EHR information, with the latter source of data reviewed for a formal GWI diagnosis or requisite evidence of appropriate signs and symptoms and chronic health conditions. Control participants will not have evidence of GWI based on self-reported survey responses. An initial pilot study of 600 Gulf War era veterans assessed feasibility of identifying and developing phenotypic data prior to initiating the fully powered study.

### 2.2. Case Definition

In a 2014 report of ongoing multisymptom illness among veterans of the Persian Gulf War, the IOM acknowledged the lack of a “gold standard” case definition for this illness [13]. Nonetheless, the report recognized that the CDC [23] and Kansas [24] definitions of GWI include the symptoms most often reported by veterans, and recommended use of these two definitions in VA research [13]. These define the GWI phenotypes to be used in the primary GWAS and G × E analyses described here.

#### 2.2.1. CDC Definition

Fukuda et al. [23] defined cases on the basis of self-reported symptoms and duration (<6 months or ≥6 months) and intensity (mild, moderate or severe) of each symptom present. Participants completed a questionnaire, which asked about 35 symptoms that had been identified during an earlier exploratory study. Following both a clinical and a statistical approach, a case was defined as a participant endorsing one or more symptoms, of at least six months duration, from two or more of the following three categories: fatigue; mood and cognition (symptoms of feeling depressed, difficulty remembering or concentrating, feeling moody, feeling anxious, trouble finding words, and difficulty sleeping); and musculoskeletal (symptoms of joint pain, joint stiffness, and muscle pain). Moreover, a case was classified as severe if each case-defining symptom was rated as severe; otherwise, the case was called mild–moderate.

The CDC definition captures the three symptoms commonly reported in the literature but is broad and inclusive (especially the mild–moderate form), resulting in a high prevalence rate [22]. In practice, it has been found to be useful in clinical settings to rule out disease [42]. This definition has been the most commonly used and is accepted internationally [15,22]. The IOM has recommended its use in clinical practice [13,14].

#### 2.2.2. Kansas Definition

Steele [24] based the criteria for defining GWI on chronic fatigue syndrome, another disease defined primarily by symptoms. Symptom groups were defined based on measures of correlation and comparisons between the deployed and the non-deployed. Participants were asked about 37 symptoms in six domains: fatigue and sleep; pain; neurologic, cognitive and mood; gastrointestinal; respiratory; and skin. For each symptom present, the duration and severity were also measured. Steele [24] also identified exclusionary conditions, including medical and psychiatric diagnoses that were not different in the deployed and non-deployed veterans, but might confound the diagnosis of GWI. Specifically, any of the following diagnoses was a reason for exclusion from consideration as a case: cancer; diabetes; heart disease; chronic infectious disease; problems resulting from postwar injuries; liver disease; lupus; multiple sclerosis; stroke; or any serous psychiatric condition, such as psychosis, bipolar disorder, or one requiring hospitalization since 1991 [32]. A case was defined as a respondent with at least one moderately severe symptom or two or more symptoms, within at least three of the six domains, and no exclusionary condition.

Overall, the Kansas definition is more restrictive than the CDC criteria and therefore identifies GWI at lower prevalence rates [22]. However, because the Kansas definition excludes veterans with common chronic diseases, it can potentially exclude GWI cases if the “comorbidities” are actually part of a Gulf-War-related ailment or they developed after onset of GWI, as part of the aging process.

The Kansas [24] and CDC [23] case definitions of Gulf War Illness are illustrated schematically in Figure 1. Symptom domains and constituent individual symptoms are specified for the two case definitions, as well as the exclusionary criteria utilized in the Kansas case definition [54].

### 2.3. Population/Sample/Recruitment

The CSP #2006 target population is the set of veterans who served in the military during the 1990–1991 Persian Gulf War era and represents a subset of MVP participants. As described elsewhere [52], the MVP sample, representing over 1 million veterans enrolled in the VHA, consists of a diverse mix of former US military service members of varying age, race/ethnicity, sex, military experience, and geographical location. As of 21 October 2020, 5,918,395 veteran users of the VHA have been invited by mail to participate in the MVP, 829,975 have enrolled, 518,610 (62.5%) have completed the baseline MVP questionnaire, 390,807 (47.1%) have completed the lifestyle survey, 351,218 (42.3%) have completed both, and 455,789 (55.0%) genotypes are available for analysis. MVP participants volunteer to participate, and are generally representative of VHA users, but differ somewhat with respect to demographic, military, and health characteristics from the population of living veterans [55].

The potential study population for CSP #2006 was identified by matching data from the Department of Defense Manpower Data Center (DMDC) with a list of MVP participants who self-reported service during the 1990–1991 Persian Gulf War era (August 1990–July 1991). Permission to use these databases was obtained in accordance with VA policies. Waivers of HIPAA authorization and informed consent were approved to generate contact information for potential study participants.

These veterans were contacted by mail and invited to participate in CSP #2006. The invitation letter was followed by a 20-page, optical-scan formatted, structured health questionnaire mailed to MVP participants who did not opt-out of the study after initial contact. The survey instrument was designed to be consistent with other GWI questionnaires used within the VA [56], which have collected service details including deployment locations and exposures, health information in the form of symptoms associated with GWI, diagnoses of specific medical and psychiatric illnesses, personal habits, and health care and hospitalization utilization.

### 2.4. Chart Reviews

To assess self-reported symptoms and disease diagnoses, and to validate case-control classification, we are conducting chart reviews for symptoms, exposures, and diagnoses recorded by clinicians in the clinical and disability examination notes. Explicit chart review rules will allow for trained reviewers to abstract relevant details from a small subset of participants. These data will be used to corroborate self-reported survey responses. Structured data, including international classification of disease (ICD) codes, will also be utilized in exploring GWI phenotypes and classifying participants by GWI case status, particularly with regard to exclusionary conditions for the Kansas case definition.

### 2.5. Phenotyping

GWI case status for the primary genomic analyses will be assigned based on self-reported responses to survey items related to symptoms and medical conditions (see appendix for survey instrument) consistent with the CDC and Kansas case definitions described above. Symptoms will be considered if onset was during or after deployment and present for more than 6 months. Symptom severity will be incorporated and allow for differentiation between CDC mild–moderate and CDC severe phenotypes. We will also explore differences between GWI defined by the Kansas case definition with the exclusionary conditions and without consideration of the exclusionary conditions.

Exposures will be derived from several survey items asking about deployment status, location in theater, and specific exposures of interest. In conjunction with self-reported branch of service, classification of exposure status will be corroborated through exploration of the self-reported responses to these items. Ambiguous or implausible patterns of response to exposure and location items will be adjudicated using rule-based algorithms derived from the published literature and input from the research team and CSP #2006 executive committee.

### 2.6. Genotyping

The collection, shipping, and storage of biospecimens, as well as specifics of the MVP genotyping microarray, have been described previously [52]. Blood samples collected from consented participants are mailed to the VA Central Biorepository in Boston, which ships them to one of two approved vendors, BioStorage Technologies, Inc., Indianapolis, IN and AKESOgen, Norcross, GA, for genotyping using a custom Affymetrix 723K chip^®^. Development of the genetic database, quality control measures, and imputation have been described [57] and are maintained by the VA’s Genomic Information System for Integrated Science (GenISIS). The current dataset includes information regarding high quality genotypes for 668,418 common and rare variants assessed in 459,777 MVP participants [57].

### 2.7. Power Calculation for Genetic Analysis

Among the veterans enrolled in MVP, 109,976 served during the 1990–1991 Persian Gulf War Era, all of whom were mailed the GW survey, and 45,044 surveys were returned. Based on these numbers and estimates of case status in similar cohorts, we expect to obtain information regarding GWI status in approximately 14,700 cases and 27,300 controls. Additional participants may be included, based on information derived from the electronic health records of all Gulf War era veterans enrolled in the MVP. Considering common alleles (effect allele frequency, EAF > 5%) with moderate effects (odds ratio, OR > 1.05), our case-control trans-ancestry meta-analysis has 80% statistical power to detect OR = 1.1 (95% confidence interval: 1.06–1.14) for EAF > 35%. However, we are underpowered to detect small genetic associations (OR ≤ 1.05, <80% statistical power). With respect to the gene–environment interaction analysis, we will conduct a multivariate gene–environment interaction analysis using the structLMM method (see description in the Data Analysis section). As previously reported [58], this approach is more powerful than a single environment, one degree of freedom fixed-effect test. The investigation of multivariate gene–environment interactions has several power advantages, including the ability to (i) identify interactions that are simultaneously driven by multiple environments; (ii) assess unobserved drivers of gene–environment interactions using combinations of multiple environmental variables as proxy; and (iii) reduce multiple testing burden due to the joint test design.

### 2.8. Data Analysis

Association and interaction analyses will be conducted using the MVP genome-wide data generated through a unified genetic quality control (QC) [57]. These genotype data were used to perform imputation with minimac3 [59], leveraging the 1000 Genomes Project reference panel [60]. For post-imputation QC, SNPs with imputation INFO scores of <0.6 or minor allele frequencies (MAF) < 0.01 will be removed from the analysis.

The MVP cohort reflects the ancestry and ethnic diversity of the US population. MVP participants reported a wide range of ancestry and ethnic backgrounds, including Hispanic European/European American, African American/Afro-Caribbean, East Asian, and others. With nearly 30% of the participants being of non-European descent, MVP is the most diverse among large biobanks [57]. To model MVP genetic diversity appropriately in our analyses, we will use a recently developed supervised learning algorithm, HARE (harmonized ancestry and race/ethnicity) [61], to define a categorical stratification variable. Leveraging HARE, we will partition the MVP multi-ethnic cohort into non-overlapping strata. Within each HARE stratum, we will use principal components (PCs) to account for the variation in admixture proportions and/or geographic cline.

Association tests will be conducted using regression models implemented in PLINK 2.0 [62] and linear mixed models implemented in BOLT-LMM [63], to investigate the four GWI-relevant case-control phenotypes (CDC, CDC-severe, Kansas symptoms and Kansas symptoms and exclusions). With respect to our primary analysis, for each SNP we will model an additive genetic effect using the standard genome-wide significance level (α) of 5 × 10^−8^. Ancestry and SNP-specific GWAS results will be databased for analysis of genetic correlation and for additional studies described below. For secondary analyses focused on multiple phenotypic outcomes, we will apply a false discovery rate correction. The association analysis will be performed in each HARE stratum, including sex, age, age-squared, and the top 10 PCs as covariates. The results generated from the HARE strata will be combined using a fixed-effects, inverse-variance weighted meta-analysis. Heterogeneity of genetic effects across strata will be evaluated in the meta-analysis. Additionally, we will also perform a cross-ancestry fine-mapping analysis to identify ancestry-specific effects.

Interaction analyses will be conducted using the recently developed StructLMM, a linear mixed-model approach to identify and characterize efficiently loci that interact with one or more environments [58]. These studies will permit us to simultaneously test the effect of multiple GWI risk factors, such as cumulative deployment time, deployment location, and self-reported exposures to pesticides, nerve gas, and pyridostigmine bromide. The StructLMM framework will identify loci with significant GWI-related interaction effects, along with evidence of heterogeneous effect sizes due to multivariate gene–environment interactions. For the loci identified, we will estimate the fraction of genetic variance explained by multivariate gene–environment interactions, and use Bayes factors (BF) between the full model and reduced models (i.e., with environmental variables removed) to identify GWI risk factors that are most relevant for the observed interactions. Similar to the association analyses described above, interaction tests will be conducted separately in each HARE stratum, including sex, age, age-squared, and the top 10 PCs as covariates, followed by meta-analysis of the stratum-specific results.

The genome-wide data generated by the association and interaction tests will provide the basis for a wide range of additional approaches, taking two general forms: (1) functional characterization of the underlying genetic architecture of GWI; and (2) categorization of the complex GWI phenotype with respect to the genetic contributions from related conditions. These analyses will require reference to external databases, such as GTEx (Genotype-Tissue Expression) that catalogs the tissue-specific regulatory potential of individual SNPs, as well as databases of additional omics features (i.e., transcriptome, epigenome, proteome, and microbiome) measured in relevant tissues, as well as comparisons to the phenotypic spectrum of loci identified in the association and interaction tests, through phenome-wide association studies (PheWAS) [64] of the MVP cohort after excluding the Persian Gulf War era veterans, UK Biobank, [65] FinnGen (FinnGen. Available online: https://www.finngen.fi/en (accessed on 21 June 2021)) [66], and other large biobanks.

To investigate the polygenicity of GWI, we will leverage established methods to model the independent associations expected across the human genome. Specifically, we will quantify the fraction of GWI phenotypic variance explained by additive effects of the variants, by using the linkage disequilibrium (LD) score regression method [67], and we will apply cross-phenotype LD score regression [68] to quantify the genetic liability of GWI that is shared (i.e., genetic correlation) with other human traits, including physical and mental disorders, biomarkers, and other relevant phenotypes. We will partition the heritability [69] across a broad set of functional annotations (e.g., cell type-specific elements) to identify the most relevant mechanisms involved in GWI pathogenesis. We will use the PrediXcan approach [70] and GTEx (Genotype-Tissue Expression) data [71] as a reference panel, to impute tissue-specific transcriptomic profiles in GWI cases and controls. The goal of these analyses is to develop risk-prediction models of genetic and genomic predictors (polygenic risk scores), combined with trans-ethnic analysis and phenotypic characterization that can be used to perform genomic structural equation modeling (SEM) [72] and Mendelian randomization (MR) [73]. These analyses at the intersection of the genomic and phenotypic complexities of GWI will enable examination of the robustness of findings and validation of analyses, as well as investigation of the mediation processes among GWI-related risk factors [74,75].

## 3. Discussion

The VA Cooperative Studies Program (CSP) #2006, “Genomics of Gulf War Illness”, represents a major milestone in the study of GWI, with the largest dataset to date of genomic and phenotypic information about veterans with GWI and veterans without GWI who served in the US military contemporaneously. The GWAS study design outlined in this paper and the primary analysis of CSP #2006 will potentially produce important findings about genetic variation associated with GWI, potential pathophysiologic underpinnings of GWI, pleiotropy with other traits, and gene × environment interactions. The methods address the challenge of conducting genomic analyses on a symptom-based condition, in the absence of a recognized diagnostic biomarker.

This project is high priority to address the concerns of Gulf War veterans and veterans of other cohorts who attribute unexplained chronic symptoms or difficult-to-diagnose illness to recalled occupational and environmental exposures. This effort represents a significant leveraging of VA research infrastructure at a scale that GWI research has not experienced to date. It holds promise for a more in-depth look at factors not previously available to scientists and clinicians and provides avenues for scientific inquiry that bypass or overcome previous challenges in understanding GWI pathophysiology.

This report focuses on the initial GWAS analysis of CSP #2006 and as work progresses the approaches employed to address the challenges of characterizing the GWI phenotype and the endophenotypes may be of more generalized relevance and applicability. There are several other chronic symptom-based conditions related to GWI, also with poorly understood pathophysiology and inadequate biomarkers of disease, such as chronic fatigue syndrome, fibromyalgia, and functional gastrointestinal disorders. Similarly, mental health conditions are mostly defined by symptoms and behaviors. The proposed study will provide an opportunity to place GWI symptoms in the context of related physical or psychiatric illnesses. Our approach to phenotyping GWI in CSP #2006 may prove transferable to other conditions and other populations with complex symptoms.

Strengths of this project include the large sample size and the richness of the self-reported data from the surveys, especially the symptom and exposure reporting. In addition, the potential for incorporating structured and free text data from the EHR is promising and will be examined in the later stages of this project. Finally, the depth and breadth of experience among the study team provides high quality technical and analytical expertise for the project.

Weaknesses of this project include the reliance on self-reported data, limitations of the two case definitions to be used in primary analyses, and a dearth of objectively determined exposure data. These challenges, not unique to this study, are inherent in current state-of-the-art GWI research. Misclassification of GWI status is a major concern and multiple sources could contribute to this problem. We will initially rely on self-reported symptoms and health conditions to determine GWI status; symptoms are subjective by nature and self-report introduces the possibility of inaccuracies, recall bias and memory problems in this aging cohort. Corroboration of self-reported responses with documentation in the medical record will provide insights into the risk of misclassification.

The association between greater age and the chronic conditions included in the original Kansas definition is of distinct concern in this aging cohort. These comorbid conditions, which likely developed after the onset of GWI in those with GWI, may result in misclassification of GWI in our sample. We have some ability to explore symptom “year of onset” and “year first told” of a diagnosis to examine the temporal relationship, although the details of these analyses will need to be developed. We will also explore variants of the recommended case definitions of GWI to assess the consistency of meaningful genomic information across phenotypes and explore possible misclassification. For example, future analyses of CSP #2006 data will explore different approaches to identifying exclusionary conditions for the Kansas GWI case definition. Another limitation of the study is the voluntary nature of participation in both MVP and CSP #2006. This self-selection bias may reduce the genetic and phenotypic variability among participants, and limit potential for discovery of genetic associations of GWI. Similarly, survival bias is a limitation that cannot be addressed with this study.

There have been very few genomic studies of GWI [28], and none as large as the one proposed for this project. In fact, with fewer than 700,000 surviving deployed Gulf War veterans, our conservatively estimated sample (50% of 45,000 survey respondents deployed to GW) represents approximately 3% of the total population of exposed individuals. Because data generated by CSP #2006 will be returned to the Million Veteran Program, findings from this study, including genotypes, will become available to future researchers, to expand upon and apply new approaches and methods as they become available. Moreover, combining genetic information with other omics information when it becomes available will contribute to future multi-omics studies of GWI. Data from multiple molecular layers may potentially allow for more accurate modeling of the complex dynamics driving GWI pathogenesis, as demonstrated in a recent analysis of potential COVID-19 treatment targets using the MVP [76].

In summary, CSP #2006 represents a rigorous study of genomic underpinnings of GWI, based on sample size and state-of-the-art genomic analysis and phenotyping. The findings could well serve as a landmark study of this disease, and the data generated will provide a rich and expansive foundation upon which to build additional analyses.

## Figures and Tables

**Figure 1 brainsci-11-00845-f001:**
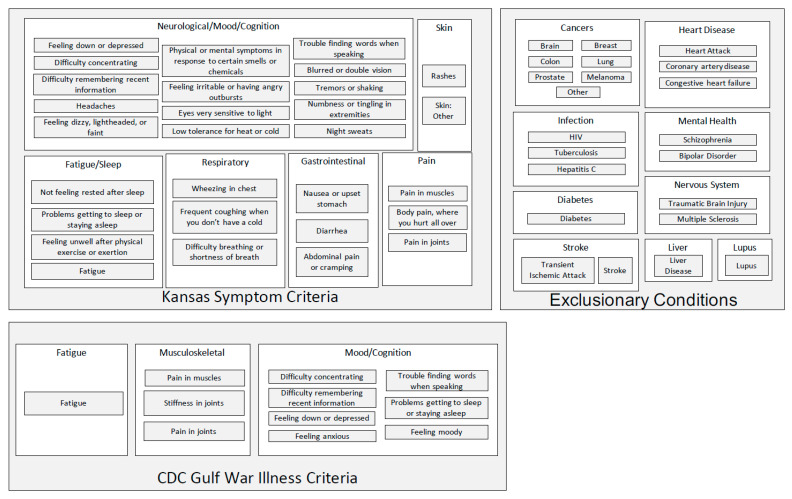
The Kansas [24] and CDC [23] case definitions of Gulf War Illness.

## Data Availability

Data and materials can be made available for review in accordance with VA CSP and MVP policy.

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
