# Peer review of "Genomics of Gulf War Illness in U.S. Veterans Who Served during the 1990–1991 Persian Gulf War: Methods and Rationale for Veterans Affairs Cooperative Study #2006"

_brainsci, 2021, doi:10.3390/brainsci11070845_

Round 1
Reviewer 1 Report
The manuscript by Radhakrishnan et al. is a methodologic description of a laudable attempt to bring highly sophisticated methods of genotypic analysis to discover DNA markers predisposing to GWI. The paper is clearly and concisely written by scientists who clearly appear to be experts in GWAS methods.
While the description of the study design, which occupies most of the text, is clear and complete, the one area inadequately treated is what is referred to in the Discussion as “weaknesses.” In the present draft this paragraph merely lists 5 serious limitations each of which could potentially render the huge investment useless. Basically, the balance between the extensive coverage of esoteric genomic analysis methods and the meager mention of weaknesses is glaring.
This inadequacy could be substantially addressed by moving this subject to become the last subsection of the Methods section, title it “Challenges to be Addressed,” and provide a full paragraph on each of the 5 challenges with a cogent plan to address each. Each could be addressed by outlining plans 1) to overcome it, 2) to assess its effect of the findings or lack of, or 3) just admitting that it could bias or obviate any findings.
To be frank, many or most ill GW veterans are angry at the VA System for denying the existence of GWI, and many of them will suspect that in this study the VA will be pulling a punch to deliberately find nothing. Many of these veterans and all of the scientific critics will know these drawbacks, and failing to emphasize how they are to be addressed will only add to the skepticism. Thus, a greater emphasis on proposed solutions would go a long way to assuage the suspicion.
Here are some thoughts on each:
1) Misclassification in the case definition. Clearly the greatest threat to success of the study is misclassification of veterans on the case definition. The CDC and Kansas case definitions are notorious for being overly sensitive and thus admitting many false-positive misclassifications. In the proposed study, this will be accentuated by the heavily volunteer nature of the sample where only 42,000 out of 660,00 deployed GWVs volunteered to be in the study, briefly addressed in the present draft. While the majority of these will be bona fide cases, a large minority will be well-meaning veterans sick with other diseases they mistakenly attribute to the Gulf War and a smaller number who are frankly dissembling for secondary gain. This would be exacerbated by following the Kansas case definition’s list of exclusions, which is likely to selectively exclude those most severely affected by GWI (a problem the paper well describes but proposes no solution for).
The paper needs to be revised to add a discussion of the extensive literature on misclassification in the case definition and its impact on the power to detect elusive genomic associations and gene-environment interactions (it potently depresses power to detect associations). That should then lead to a more complete description of the already stated plan to refine the case definition for the study but in the context of reducing misclassification. The stated plan to revise the case definition to fit genomic associations better appears circular; it might better be replaced by some type of principle components approach to find a case definition unique to ill GW veterans, see the present references 24, 34, 45 and 49 as well as Kang et al. (Arch Env Health 2002;57:61-68).
2) "A dearth of objectively determined exposure data." Over the years, this criticism has been used incessantly to disqualify studies that found epidemiologic associations with environmental risk factors, so it must be overcome to make the proposed study useful. In the current draft, there is great emphasis on consulting electronic medical records, evaluating ICD diagnosis codes, and the like. Since VA physicians have long been trained to deny GWI, it is hard to see how these sources will be useful to refining a case definition. On the other hand, the paper would be enhanced by a short review of prior studies where various self-reported environmental exposures were evaluated for repeatability or were found to be associated with objective outcome measures, or the like. More thought needs to be given to creative ways to solve this potentially devastating problem.
3) Use of the Kansas case definition. The present draft discusses the problem of exclusions in the Kansas definition, but does not reach a conclusion on what to do about it. It is well known that the chronic illnesses that constitute these exclusions increase with age and disproportionately affect those with GWI who are made sedentary by their condition. Over time this exclusionary principle has disqualified a disproportionately increasing share of veterans with GWI. This will greatly reduce analytic power and is likely to exclude those with the most crippling forms of GWI; such a selection bias may reduce the study to one of only the mild forms of GWI and thus trivialize the findings. Most recent studies who say they use the Kansas definition also specify that either they modify the list of exclusions or drop it altogether. A decision on what to do with the Kansas exclusions should be made and stated in the paper.
Other issues:
There appears to be a disconnect between the first clause of the last sentence in the abstract (“a more reliable and robust definition of GWI will benefit research . . .”) and lines 18 and 19 on p. 8 (“. . . but validating a new case definition is beyond the scope of this project”).
Here the authors appear to be treading on dangerous ground, intending to come up with a new case definition that should become widely used but without putting in the effort to validate it. Prior GWI case definitions have been developed on a few hundred veterans and validations on hundreds or a few thousand. This study will have as many as 14,700 GWI cases and 27,300 GWV controls! Why not propose to divide these large samples into a Developmental Sample and a Validation Sample, as is done in other fields the authors must be familiar with (a regular feature of GWAS). Develop the new case definition on one and show its fit to the other. The resulting samples will still be far larger than required for a nice study that might come out with a new validated definition, assuming of course that the problems listed above are solved and a study of DNA can identify a new case definition. Given that this is a central enough goal of the study to list in the conclusion of the abstract and that a validation step is clearly practical, this should be proposed.
If manuscript length is a limitation, the above proposed discussion of how the authors propose to resolve the obvious methodologic problems would be far more interesting to the large body of skeptical scientists and veterans than the esoteric discussion of advanced genomic analytic approaches which presently occupy a disproportionately large amount of space in the paper.
Reviewer 2 Report
This is a multicenter interdisciplinary large scale study of genetic susceptibility to GWI. The investigators take advantage of existing infrastructures and resources including the VA Million Veteran Program focusing on genetic analysis and the VA electronic medical records. Advanced statistical and machine learning approaches will be used for the data analysis. One of the goals of the study will be also be to study gene-environment interactions in the GWI cohort and shed light to the underlying disease mechanisms.
They will use a case control study design and participants will be derived from those who served during the Persian Gulf War. While there is a discussion of the CDC and Kansas definitions of GWI, the authors do not clearly state which exact criteria will use to select the GWI cases. Information about symptoms and potential exposures will be gathered from mailed questionnaires. Since more than 30 years have passed form the actual exposure, it is not clear whether symptoms will be considered if current or if they occurred at any time in the past. Along the same lines, review of medical records is a time consuming task that needs vast resources for the number of patients to be included in VA cooperative study #2006. Is GWI included as a diagnosis in those records? What is the period that this review will cover? Again, specific inclusion and exclusion criteria and primary and secondary endpoints need to be described a priori. Patient grouping according to their genetic makeup is a plus of the study design; however, considerations of how this HARE stratification might affect the current power calculation should be included in the analysis plan. In its current form, the study is 80% powered for OR=1.1. Confidence intervals for OR should also be provided. Since it is most likely that the variability in the phenotype will be explained by more than one gene, a predetermined number of genes that will be analyzed together and included in the model should be specified.
It is mentioned that
There is a plan to interrogate additional OMICS-databases for GWI. Those resources, if available, should be specified or that attempt can be included in the discussion as future plans
The authors should be congratulated for including an accurate evaluation of the strengths and limitations of their study
Overall, the genetic analysis studies will become a landmark in the GWI study and expected to illuminate several critical aspects of disease-related phenotypes
Reviewer 3 Report
Pros/strengths: This study utilizes a large well characterized sample. The potential to determine genetic/epigenetic susceptibility in some GW veterans will assist in sub group analyses of pathophysiology and treatment studies and may help with development of biomarkers. Limitations: Misclassification of GWI cases is concerning due to lack of reliable case definition, which for this aging cohort is an acknowledged weakness, as per Gifford et al, Gulf War illness in the Gulf War Era Cohort and Biorepository: The Kansas and Centers for Disease Control definitions, 2021, https://doi.org/10.1016/j.lfs.2021.119454 The authors propose to assess this by exploring "variants of the case definition of GWI in efforts to utilize a phenotype that correlates with meaningful genomic information". This is a difficult problem with no easy solution, but generating more data in this study should be helpful.Author Response
Please find the attachment.
